# Low Serum Vitamin D Status Is Associated with Incident Alzheimer’s Dementia in the Oldest Old

**DOI:** 10.3390/nu15010061

**Published:** 2022-12-23

**Authors:** Debora Melo van Lent, Sarah Egert, Steffen Wolfsgruber, Luca Kleineidam, Leonie Weinhold, Holger Wagner-Thelen, Birgit Stoffel-Wagner, Horst Bickel, Birgitt Wiese, Siegfried Weyerer, Michael Pentzek, Frank Jessen, Matthias Schmid, Wolfgang Maier, Martin Scherer, Steffi G. Riedel-Heller, Alfredo Ramirez, Michael Wagner

**Affiliations:** 1German Center for Neurodegenerative Diseases (DZNE), 53127 Bonn, Germany; 2Glenn Biggs Institute for Alzheimer’s and Neurodegenerative Diseases, University of Texas Health Science Center, San Antonio, TX 78229, USA; 3Institute of Nutrition and Food Sciences, Nutritional Physiology, University of Bonn, 53115 Bonn, Germany; 4Department of Neurodegenerative Diseases and Geriatric Psychiatry, University of Bonn, 53105 Bonn, Germany; 5Department of Medical Biometry, Informatics and Epidemiology, University Hospital Bonn, 53105 Bonn, Germany; 6Department of Psychiatry, Medical Faculty, University of Cologne, 50924 Cologne, Germany; 7Institute of Clinical Chemistry and Clinical Pharmacology, University of Bonn, 53127 Bonn, Germany; 8Department of Psychiatry, Technical University of Munich, 81675 Munich, Germany; 9WG Medical Statistics and IT-Infrastructure, Institute of General Practice, Hannover Medical School, 30625 Hannover, Germany; 10Central Institute of Mental Health, Medical Faculty Mannheim, Heidelberg University, 68159 Mannheim, Germany; 11Institute of General Practice, Medical Faculty, Heinrich-Heine-University, 40227 Dusseldorf, Germany; 12Department of Primary Medical Care, Center for Psychosocial Medicine, University Medical Center Hamburg-Eppendorf, 20246 Hamburg, Germany; 13Institute of Social Medicine, Occupational Health and Public Health, University of Leipzig, 01403 Leipzig, Germany

**Keywords:** vitamins, vitamin D, beta-carotene, apolipoprotein E ε4, dementia, Alzheimer’s disease dementia, elderly

## Abstract

Background. Vitamins A, D and E and beta-carotene may have a protective function for cognitive health, due to their antioxidant capacities. Methods. We analyzed data from 1334 non-demented participants (mean age 84 years) from the AgeCoDe study, a prospective multicenter-cohort of elderly general-practitioner patients in Germany, of whom *n* = 250 developed all-cause dementia and *n* = 209 developed Alzheimer’s dementia (AD) during 7 years of follow-up. We examined whether concentrations of vitamins A (retinol), D (25-hydroxycholecalciferol) and E (alpha-tocopherol) and beta-carotene, would be associated with incident (AD) dementia. Results. In our sample, 33.7% had optimum vitamin D concentrations (≥50 nmol/L). Higher concentrations of vitamin D were associated with lower incidence of all-cause dementia and AD (HR 0.99 (95%CI 0.98; 0.99); HR0.99 (95%CI 0.98; 0.99), respectively). In particular, subjects with vitamin D deficiency (25.3%, <25 nmol/L) were at increased risk for all-cause dementia and AD (HR1.91 (95%CI 1.30; 2.81); HR2.28 (95%CI 1.47; 3.53), respectively). Vitamins A and E and beta-carotene were unrelated to (AD) dementia. Conclusions. Vitamin D deficiency increased the risk to develop (AD) dementia. Our study supports the advice for monitoring vitamin D status in the elderly and vitamin D supplementation in those with vitamin D deficiency. We observed no relationships between the other vitamins with incident (AD) dementia, which is in line with previous observational studies.

## 1. Introduction

The worldwide number of elderly and oldest old (85+ years) is rising and simultaneously the number of people with dementia [1,2]. In addition, the prevalence of vitamin deficiencies is higher among the elderly than among middle-aged adults [3,4], making this age group more prone to chronic non-communicable diseases, including dementia. The research focus should, therefore, go to modifiable risk factors, such as diet, to counteract this trend.

Fat-soluble vitamins with anti-oxidative properties have been linked with cognitive health outcomes and may help protect against dementia risk factors [5]. What are the main functions of these promising vitamins in the brain? Vitamin D contributes to neurotransmitter release, functions as a neuroprotector and fights against oxidative stress and pro-inflammatory agents [5]. Vitamin A and beta-carotene (pro-vitamin A) function as antioxidants that tend to scavenge lipid peroxidation and neurotransmitter release [5,6]. In addition, vitamin E is the main lipid-soluble antioxidant and its most common form is alpha-tocopherol [5]. Compared to the different vitamin E forms, alpha-tocopherol has the highest bioavailability. However, it has been shown that tocotrienols have stronger antioxidant properties [7,8]. In the brain, vitamin E acts as a cellular membrane protector and neuroprotector by counteracting oxidative stress and fighting against Aß free radicals [9,10]. To date, results from randomized control trials (RCTs) which investigated the effects of these dietary antioxidants on cognitive decline and the risk of dementia have not shown conclusive protective effects [11]. In contrast, in observational research, vitamin D seems to be the most promising fat-soluble vitamin which may decrease the risk for dementia [12]. Although less widely studied, promising findings have also been reported (in mainly cross-sectional studies) for a relationship between higher concentrations of vitamin E (i.e., alpha-tocopherol) and a decreased risk for (incident) dementia [13,14], but not in all studies [15,16,17]. Further, cross-sectional studies have reported relationships between low concentrations of vitamin A [6] and beta-carotene [18] and increased odds of dementia. However, observational studies investigating the longitudinal relationships between vitamin A and beta-carotene concentrations and incident dementia are lacking.

We examined the relations between vitamins A, D and E and beta-carotene with incident all-cause dementia and AD among the oldest old participants of the AgeCoDe multicenter cohort study in Germany.

## 2. Materials and Methods

### 2.1. Study Design

The current data are from the German Study on Ageing, Cognition and Dementia in Primary Care Patients (AgeCoDe) and Needs, health service use, costs and health-related quality of life in a large sample of oldest-old primary care patients aged 85+ (AgeQualiDe). The study is a multicenter and general practitioner (GP) registry-based prospective cohort study on the early detection and prediction of dementia, starting in 2003. The AgeCoDe baseline visit selection and sampling of the participants have been described previously [19,20]. In brief, enrolled participants were primary care patients aged 75 years or older, being non-demented at baseline, who had at least one personal contact with their GP during the past year, and who were living in the urban areas of the six German cities Bonn, Düsseldorf, Hamburg, Leipzig, Mannheim or Munich [19]. The recruitment was conducted by 138 GPs connected to the respective study site. At baseline, 3327 participants consented to enrollment in the study. Eight follow-up (FU) visits (with an 18-month interval between each FU) were completed up to the time of the present study. For participants who could not be interviewed personally at any FU visit, informant-based information was obtained. In such cases, participants were excluded from further FUs.

The study was approved by the local ethics committees of the six participating centers and all participants gave their written informed consent to the study.

### 2.2. Study Participants

For the present study, we used data from participants who had available FU data from FU-3 (time of blood sampling for dietary antioxidants) until FU-8. We excluded participants who did not meet the initial AgeCoDe study baseline visit inclusion criteria (*n* = 117), died (*n* = 511) or dropped out before FU-3 (*n* = 763), resulting in a sample of 1936 participants at FU-3. From the 1936 participants at FU-3, we excluded participants for whom vitamin A, vitamin D, vitamin E, beta-carotene and/or cognitive test data were not available (*n* = 553). We also excluded 108 participants with prevalent dementia at FU-3 or missing incidence dementia data, resulting in a sample consisting of *n* = 1334 participants. Details on the exclusion of participants from the present study are presented in Figure 1.

### 2.3. Blood Concentrations of Vitamins A, D, and E, Beta-Carotene, Creatinine, Total Cholesterol and Triglycerides

Blood was collected by participants’ GPs in tubes (500 µL aliquots) with ethylenediaminetetraacetic acid (EDTA) and without anticoagulant, protected from light and heat for transportation and stored at −80 °C. Retinol, α-tocopherol and β-carotene were determined by high-performance liquid chromatography (HPLC). Therefore, 500 µL plasma was deproteinized by the addition of ethanol. The fat-soluble vitamins were extracted according to the modified protocol of Erhardt et al. [21] using apocarotenal as the internal standard, Nucleosil 100-5 CN (Macherey-Nagel, Düren, Germany) as the column, and a solution of 98.5% hexane and 1.5% isopropanol as mobile phase. The HPLC system consisted of the following components: intelligent-pump Prostar (Varian, Darmstadt, Germany), UV-VIS Detector Smartline 2600 (Knauer, Berlin, Germany), injection valve (Sykam, Fürstenfeldbruck, Germany), HPLC software Chromstar (SCAP, Weyhe-Leeste, Germany). The flow rate was maintained at 1.5 mL/min. Retinol was detected at 325 nm (CV 0.9%), α-tocopherol at 292 nm (CV 0.8%) and β-carotene at 450 nm (CV 0.9%), respectively. All measurements were performed from frozen plasma samples and in duplicate.

Vitamin D, creatinine, total cholesterol and triglycerides were serum samples measured during routine diagnostics of the central laboratory at the University Clinics Bonn, Germany, which is accredited according to the international standard DIN EN ISO 15189 as a part of clinical examinations. Internal and external quality control accuracy specifications were fulfilled according to the guidelines of the German Federal Medical Society (RiliBÄK) [22]. Serum vitamin D (25-hydroxycholecalciferol) was determined by fully automated commercially available enzyme-linked immunoassay kits with a sample volume used of 10 µL according to the manufacturer’s instructions and quality controls on an iSYS™ immunoanalyzer (IDS, Frankfurt/Main, Germany). The coefficients of variation for intra-assay and inter-assay precision were 3.34% and 4.96%. Serum creatinine, total cholesterol, and triglycerides were analyzed by fully automated assays on a cobas c702 analyzer (Roche Diagnostics, Mannheim, Germany) with a sample volume used of 10 µL for the creatinine assay and 2 µL each for the total cholesterol and triglyceride assays according to the manufacturer´s instructions (Roche Diagnostics). The coefficients of variation for intra-assay and inter-assay precision were 2.53% and 3.42% for creatinine, 0.82% and 1.68% for total cholesterol, and 0.91% and 1.84% for triglycerides, respectively.

### 2.4. Assessment and Diagnosis of Dementia

Dementia was diagnosed by consensus of the interviewing investigator and an experienced geriatrician or geriatric psychiatrist according to DSM-IV and International Classification of Diseases (ICD-10) criteria that are implemented as a diagnostic algorithm in the structured interview for the diagnosis of dementia (SIDAM) [23,24]. This algorithm comprises cognitive impairment, as defined by the total SIDAM cognitive score (SISCO, scoring 0–55 with a higher score indicating a better performance as the sum of the Mini-Mental State Examination (MMSE) score (0–30) and 25 additional items) and impairment of activities of daily living (ADL) as defined by a score of at least two points on the SIDAM-ADL-scale. For dementia, the etiological diagnosis of AD was established according to the National Institute of Neurological and Communicative Disorders and Stroke and the Alzheimer’s Disease and Related Disorders Association (NINCDS-ADRDA) criteria for probable AD [25]. For the diagnosis of vascular dementia, that is, in case of evidence for cerebrovascular events (Hachinski–Rosen Scale, medical history) and a temporal relationship between the cerebrovascular event and the occurrence of cognitive decline, the National Institute of Neurological Disorders and Stroke and Association Internationale pour la Recherché et l´Enseignement en Neurosciences (NINDS-AIREN) criteria were used [26]. Mixed dementia was diagnosed in cases of cerebrovascular events without a temporal relationship to cognitive decline. Dementia diagnosis in participants who were not personally interviewed was based on the GDS [27] and the Blessed Dementia Rating scale (BDR) [28]. A score of 4 or higher on the Global Deterioration Scale was used as the criterion for the dementia diagnosis. In these cases, an etiological diagnosis was established if the information provided was sufficient to judge etiology according to the above-named criteria. In addition, for statistical analyses, AD and mixed dementia were combined into one AD group.

### 2.5. Confounders

Assessments were performed by trained investigators (physicians, psychologists, gerontologists) at participants’ homes and included structured clinical interviews comprising clinical, sociodemographic and anthropometric information, neuropsychological tests, current physical and mental health, and psychosocial and lifestyle factors.

Education was classified into three levels (low, middle, and high) based on the Comparative Analysis of Social Mobility in Industrial Nations (CASMIN) classification system [29]. For genetic and blood biomarker analyses, blood samples were drawn from each participant at the attending GP practice. APOE genotyping was performed according to standard procedures [30]. Participants were grouped into those with at least one APOE ε4 allele and those without an ε4 allele. Body height and weight were measured at FU-3. BMI was calculated based on the weight in kilograms (kg) divided by height squared. Smoking status was assessed at the baseline visit and used as a proxy for FU-3. Smoking was divided into three categories as never smoker, a former smoker and a current smoker. Assessment of physical activity was conducted at FU-3. We constructed a modified physical activity score based on Verghese et al. [31]. Participants reported the frequency of usual engagement in each of the six physical activities: bicycling, walking, swimming, gymnastics, chores/gardening and a category of other physical leisure activities (e.g., bowling, jogging, or golfing) using five possible options to answer: (1) “every day”; (2) “several times per week”; (3) “once a week”; (4) “less than once a week” and (5) “never”. For the present study, the five frequency categories were collapsed into two categories whether the participant usually engaged in one of the six activities (every day and once a week = 1) or not (less than once a week and never = 0). For each participant, these values (0 or 1) were summed up across the six activities to a total physical activity score (scoring 0–6). Serum concentrations of total cholesterol, triglycerides and creatinine (FU-3) were determined by the p- and ecobas^®^ modular platform from the manufacturer Roche. Blood pressure was assessed at FU-3 by the participants’ GP. Vitamin (D) supplement intake was based on self-report. Seasonality was calculated using the date of blood sampling. Cognitive decline was calculated using the Consortium to Establish a Registry for Alzheimer’s Disease (CERAD) neuropsychological assessment battery (we applied the validated German version [32]) 10-item Word List Delayed Recall subtest (scoring 0–10; higher scores indicated a better memory performance). We subtracted the cognitive test data of FU-3 from the test data from the baseline measurement.

### 2.6. Statistical Analysis

Cox proportional hazard models (hazard ratios; HR, 95% confidence interval; CI) were used to investigate the longitudinal associations between vitamins A, D and E and beta-carotene and the incidence of AD or all-cause dementia. We tested the proportional hazard assumption using the statistical computing software R (cox.zph), and the proportional hazard assumption was satisfied for all. Further, we investigated vitamin A, vitamin E and beta-carotene as continuous variables. Vitamin D was investigated as a continuous variable and categorized according to clinical cut-off values (<25 nmol/L = deficiency; ≥25–<50 nmol/L = insufficiency; ≥50 nmol/L = sufficiency), taking sufficient vitamin D concentrations as the reference category [33]. Confounders were selected based on published literature. Nine confounders (BMI, APOE ε4, creatinine, total cholesterol, triglycerides, physical activity, vitamin supplement intake, and systolic and diastolic blood pressure) contained missing values (Appendix A). The percentages of missing values ranged from 1.6% (vitamin supplement intake) to 20.1% (triglycerides). To account for potential attrition bias, multiple imputations were used by creating ten different possible copies of the original dataset, in which the missing values were substituted by imputed values (Appendix A). These imputed values were calculated from their predictive distribution based on the observed data [34]. The combined results of the created datasets (*n* = 10) were then pooled in a separate pooled dataset to account for the uncertainty about the missing values. A description of the procedure is reported in Appendix A. All models were run using imputed data.

Models were adjusted for social-demographic factors: age, sex, education and APOE ε4 status (Model 1); additionally to Model 1 covariates, for the lifestyle factors BMI, smoking status and physical activity, creatinine, total cholesterol, triglycerides, systolic blood pressure, diastolic blood pressure, vitamin (D) supplement intake and cognitive decline before FU-3 (Model 2). The inclusion of this cognitive decline measure as a covariate is one way to account for reverse causality, which might be present if incipient dementia (reflected in ongoing cognitive decline) changes dietary habits and vitamin concentrations even before a diagnosis is made. We investigated interactions between sex and APOE ε4 status and the vitamins in Model 2 (*p* < 0.10). When significant, stratified analyses were performed according to sex status (men or women) or APOE ε4 status (carrier or non-carrier). Further, as a sensitivity analysis, we investigated the relationships between vitamin D concentrations and vascular dementia and all-cause dementia. A *p*-value < 0.05 was considered statistically significant. International Business Machines Corporation (IBM) Statistical Package for the Social Sciences (SPSS) statistics for Windows (release 23) was used to perform the analyses.

## 3. Results

Table 1 details the study population characteristics, subdivided by study sample. The mean age was 84 years, women were slightly more represented than men and the mean BMI was 25.9 kg/m^2^. The APOE ε4 allele was present in 19.3% of the participants. Mean vitamin A and vitamin E concentrations were 0.54 mg/L and 15.73 mg/L, respectively. Moreover, median beta-carotene and vitamin D concentrations were 0.32 mg/L and 37 nmol/L, respectively. Over approximately seven years of follow-up, 250 participants developed all-cause dementia, of whom 209 participants developed AD and 41 participants developed vascular dementia.

### 3.1. Longitudinal Associations between Vitamins and Beta-Carotene and Incidence of All-Cause Dementia

Higher vitamin D concentrations were significantly associated with a lower incidence of all-cause dementia (Model 1: HR = 0.99, 95% CI 0.98, 0.99) (Table 2). After additional adjustments for lifestyle factors and cardiometabolic health risk factors, the relationship remained (Model 2: HR = 0.99, 95% CI 0.98, 0.99). In addition, vitamin D was also examined categorically by using the existing reference value (i.e., serum concentration ≥ 50 nmol/L 25-hydroxyvitamin D) as a reference. When we analyzed vitamin D concentrations categorically (deficiency or insufficiency vs. sufficiency of vitamin D concentrations) we observed that vitamin D concentrations <25 nmol/L vs. ≥50 nmol/L were associated with an increased risk for incident all-cause dementia (Model 1: HR = 1.75, 95% CI 1.28, 2.40; Model 2: HR = 1.91, 95% CI 1.30, 2.81). Vitamin D status among individuals who developed all-cause dementia and individuals who did not are presented in Appendix A. For vitamins A, E, and beta-carotene no significant associations with all-cause dementia were observed.

### 3.2. Longitudinal Associations between Vitamins and Beta-Carotene and Incident AD

Higher vitamin D concentrations were significantly associated with a lower incidence of AD (Model 1: HR = 0.98, 95% CI 0.98, 0.99) (Table 3). The association remained significant after a full adjustment of confounding factors (Model 2: HR = 0.98, 95% CI 0.98, 0.99). We observed that vitamin D concentrations < 25 nmol/L and ≥25–<50 nmol/L vs. ≥50 nmol/L were associated with a higher incidence of AD (Model 2: HR = 2.28 95% CI 1.47, 3.53; HR = 1.52, 95% CI 1.04, 2.22, respectively) (Table 3). Further, we observed no significant associations between vitamin A, E, or beta-carotene and the incidence of AD.

### 3.3. Effect Modification

We observed a significant interaction between beta-carotene and sex for incident all-cause dementia and incident AD (P*_interaction_* = 0.03 and 0.01, respectively). Analyses stratified for sex revealed that beta-carotene was not significantly associated with incident all-cause dementia and incident AD among men or women (Table 4 and Table 5).

### 3.4. Longitudinal Associations between Vitamins and Incidence of Vascular Dementia

Higher vitamin A, D, E and beta-carotene concentrations were not significantly associated with incident vascular dementia. Additionally, vitamin D concentrations <25 nmol/L vs. ≥50 nmol/L were not associated with an increased risk for incident vascular dementia (Appendix A). However, vitamin D concentrations ≥25–<50 nmol/L vs. ≥50 nmol/L were associated with a decreased risk for incident vascular dementia (Model 1: HR = 0.30, 95%CI 0.13, 0.71; Model 2: HR = 0.31, 95% CI 0.12, 0.78).

## 4. Discussion

In a prospective multicenter cohort study, we investigated the associations between vitamins A (retinol), D (25-hydroxyvitamin D) and E (alpha-tocopherol) and beta-carotene and incident all-cause dementia and AD. We observed longitudinal associations between higher serum vitamin D concentrations and decreased incident all-cause dementia and incident AD in the oldest old. Sensitivity analyses revealed that higher serum vitamin D concentrations were not associated with incident vascular dementia.

### 4.1. Findings in Other Studies

Vitamin D—our finding that low status of vitamin D is associated with an increased incidence of all-cause dementia and incident AD is consistent with previous population-based longitudinal studies [35,36,37,38,39]. The Cardiovascular Health Study (CHS) reported that vitamin D deficiency and insufficiency vs. vitamin D sufficiency were associated with an increased incidence of all-cause dementia and AD [35]. Likewise, the Three-City (3C) study also found that vitamin D deficiency and insufficiency vs. vitamin D sufficiency were associated with an increased incidence of all-cause dementia and AD [39]. In addition, results from the Copenhagen City Heart (CCH) study revealed that vitamin D deficiency vs. vitamin D sufficiency concentrations were associated with an increased incidence of all-cause dementia and AD. However, vitamin D insufficiency was only associated with an increased risk for all-cause dementia, and no relationships were observed between vitamin D and incident vascular dementia [36]. Further, the Rotterdam study (RS) reported a relationship between higher vitamin D concentrations and increased risk for incident AD and all-cause dementia, but results from cut-off analyses were not significant [40]. Moreover, results from the Mini–Finland health survey (MFhes) showed that vitamin D insufficiency and sufficiency vs. deficiency were associated with a decreased risk for all-cause dementia among women but not among men [38]. These findings are in concert with the Rotterdam study, which found stronger (significant) effect estimates among women compared to men [38]. In contrast, the EPIDemiologie de l’OStéoporose (EPIDOS)-Toulouse study (*n* = 40) revealed that vitamin D deficiency at baseline was associated with the onset of non-Alzheimer dementias, but not with AD [37]. Additionally, five studies failed to observe any associations between vitamin D and incident (AD) dementia [37,41,42,43,44]. Explanations for the discrepancies in findings may be possible selection bias, power, age and vitamin D status. The sample sizes of the EPIDOS-Toulouse study and the Atherosclerosis Risk in Communities (ARIC) study were smaller compared to the aforementioned studies, which may have introduced selection bias and/or a lack of power to find a relationship with (incident) AD and all-cause dementia [37,41,42]. Two studies [42,43] were performed with slightly younger participants and very few participants in a third study [44] had vitamin D concentrations below 20 ng/mL. At last, our finding for vitamin D insufficiency and a decreased incidence of vascular dementia has not been shown in previous studies. It may be that insufficient vitamin D still contributes to a decrease in the risk for incident vascular dementia. However, the CCH study investigated vitamin D in relation to incident vascular dementia, where effect estimates pointed to the expected (i.e., increased incident) direction, but the associations were not significant [36]. We encourage future studies to examine incident vascular dementia along with AD and all-cause dementia. To date, most results of RCTs have failed to show an effect of vitamin D supplementation, due to the “non optimal” (protective) status of vitamin D at the study baseline, small sample sizes and short study durations [12]. Vitamin D is involved in several brain functions such as counteracting oxidative stress. Oxidative stress, which is an imbalance between antioxidant capacity and ROS, occurs when intracellular Aß peptide in the brain binds to mitochondrial membranes, consequently interfering with the normal electron flow through the respiratory chain causing the levels of ROS to increase. Furthermore, the imbalance induces cellular dysfunction and degeneration in the brain [45,46]. Other brain functions vitamin D is involved in are the inflammation pathway, the release of neurotransmitters, calcium homeostasis, beta-amyloid (Aß) deposition and the modulation of the immune system [47,48]. The vitamin D receptor, which functions as a gene regulator, is highly expressed in several areas of the brain: the cortex, hippocampus, dopamine neurons and the brain nucleus [49,50,51,52]. Changes in vitamin D-related gene expression might induce aging and neurodegeneration [53]. It is hypothesized that at first, Aß is able to obstruct the use of vitamin D, causing “inefficient utilization of vitamin D”. Secondly, vitamin D deficiency may induce an accumulation of Aß due to an imbalance of calcium homeostasis, or oxidative stress [53].

Vitamin A—We observed no association between serum vitamin A and incident dementia. Our finding is in line with single (cross-sectional) observational studies which found no significant relationships between vitamin A and (AD) dementia, global cognition and cognitive decline [54,55,56]. In contrast, a meta-analysis of case-control studies found significantly lower levels of vitamin A in (AD) dementia participants compared to controls [6]. However, adjustments were only made for age. Future studies are encouraged to investigate vitamin A in relation to incident (AD) dementia.

Beta-carotene—in concert with our findings, the majority of (cross-sectional) studies found no association between beta-carotene and (AD) dementia, global cognition, or cognitive decline [54,55,57,58,59,60]. In contrast, others did find significantly lower concentrations of beta-carotene in dementia cases [18,61,62,63]. However, these studies were either small or minimally adjusted for confounders. To the best of our knowledge, no meta-analysis and no longitudinal studies have been carried out to investigate the relationship between beta-carotene and (AD) dementia.

Vitamin E—Our finding that vitamin E is not associated with the risk for AD or all-cause dementia is in line with a small number of previous observational studies [13,15,16,64]. Results of these studies revealed that total tocopherol and tocotrienol, rather than alpha-tocopherol alone, are associated with cognitive decline and incident AD. To date, no meta-analysis including longitudinal studies has been carried out. Further, RCTs are few and have shown mixed results [65,66,67]. For example, one RCT observed a slowing functional decline in AD patients after receiving 2000 IU/d of vitamin E (alpha-tocopherol) over 2.27 years of FU [68]. While another trial reported lower and higher levels of oxidative stress in AD participants [69].

### 4.2. Strengths and Limitations

An important strength of our study is the prospective study design in the oldest old. Furthermore, we were able to investigate blood concentrations of vitamins, which give a better reflection of their status in the human body as compared to dietary intake. In addition, we were able to adjust for a wide range of confounders. Particularly, we were able to address reverse causality by adjusting the analyses for cognitive status at baseline. We also recognize that our study has its limitations. A single measurement of vitamin concentrations in serum or plasma may not reflect the vitamin status of the body [16,48]. However, longitudinal studies have reported the longitudinal stability of plasma vitamin E and vitamin D [48,70,71]. Furthermore, while we did not have dietary vitamin supplement data on the specific vitamins, we were able to adjust the associations for total vitamin supplement intake. In addition, we did not have data on sun exposure (Ultraviolet-B radiation stimulates cutaneous vitamin D synthesis), but we were able to adjust for the month of blood assessment. Our study has an observational study design and despite the fact that we adjusted for multiple confounders, residual confounders might still be present. Additionally, a causal relationship cannot be established but needs to be investigated further in intervention studies. We cannot completely rule out selection bias, as we selected our samples based on data on vitamins and cognitive health. However, we were able to investigate vitamins A, D and E and beta-carotene in a large sample.

## 5. Conclusions

Our study strongly supports the role of vitamin D for incident all-cause dementia and AD in the oldest old. To date, key findings for vitamin D have been reported in population-based studies. Therefore, large RCTs are warranted. Further, our findings support the recommendation of vitamin D supplementation in those with vitamin D deficiency. We observed no relationships between the other vitamins and incident AD, all-cause dementia and vascular dementia, which is in line with previous observational studies.

## Figures and Tables

**Figure 1 nutrients-15-00061-f001:**
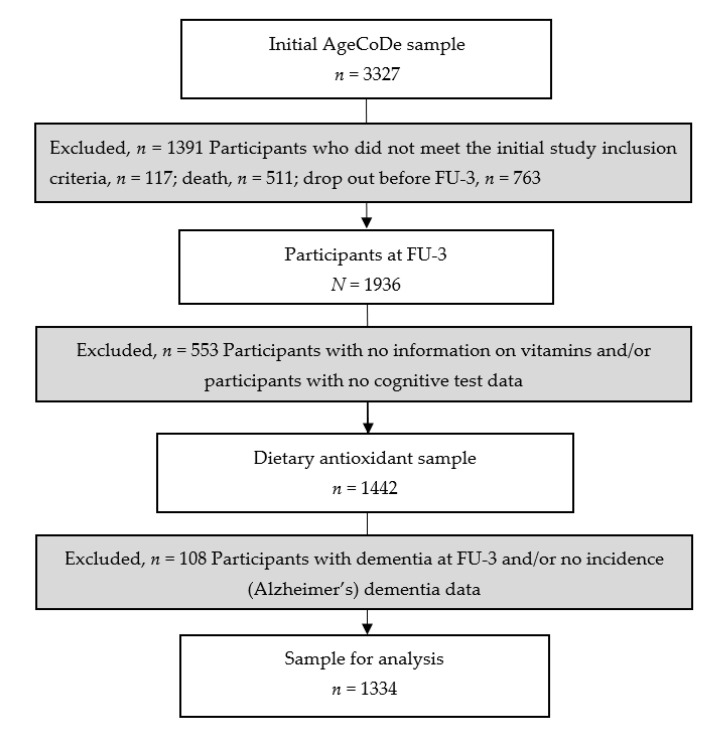
Flow chart of participants included in the analyses. Abbreviations: AgeCoDe, German Study on Ageing, Cognition and Dementia in Primary Care Patients; n, number; FU, follow up.

**Table 1 nutrients-15-00061-t001:** Characteristics of study participants.

Characteristics	Main Sample
	*n* = 1334
Age (years)	84 ± 3
Female (*n*, %)	861 (64.5)
BMI (kg/m^2^)	25.9 ± 3.7
*APOE* ε4 allele (*n*, %)	257 (19.3)
Time to develop all-cause dementia (years)	3 ± 2
Time to censoring (years)	5 ± 2
Serum/plasma analyses	
Vitamin A (mg/L)	0.54 ± 0.23
Beta-carotene (mg/L)	0.32 (IQR: 0.22–0.48)
Vitamin D (nmol/L)	37.0 (IQR: 24.8–58.3)
Vitamin E (mg/L)	15.73 ± 6.33
Creatinine (mg/dL)	1.00 (IQR: 0.83–1.22)
Total cholesterol (g/L)	2.21 ± 0.48
Triglycerides (g/L)	1.11 (IQR: 0.87–1.49)
Education (*n*, %)	
Lower	779 (58.4)
Middle	396 (29.7)
High	159 (11.9)
Physical activity (*n*, %)	
Low (0–1)	114 (8.5)
Middle (2)	906 (67.9)
High (3–5)	314 (23.5)
Smoking (*n*, %)	
Never	681 (51.0)
Past	569 (42.7)
Current	84 (6.3)
Vitamin supplement intake (*n*, %)	84 (6.3)
Systolic blood pressure (mmHg)	136 ± 16
Diastolic blood pressure (mmHg)	80 (IQR: 70–80)
ACE inhibitors usage (*n*, %)	47 (3.5)
Calcium channel blockers usage (*n*, %)	5 (0.4)
Ginkgo biloba usage (*n*, %)	10 (0.8)
Laxative usage (*n*, %)	9 (0.9)

Based on imputed data. Values are means ± SD, numbers (valid percentages), or medians (interquartile range). Abbreviations: AgeCoDe, German Study on Ageing, Cognition and Dementia in Primary Care Patients; *APOE* ɛ4, apolipoprotein ɛ4; *n*, number; mmHg, millimeters of mercury.

**Table 2 nutrients-15-00061-t002:** Longitudinal associations between blood (serum/plasma) concentrations of vitamins A, D and E and beta-carotene and incident all-cause dementia over a 7-year follow-up period (*n* = 1334).

	Incident All-Cause Dementia (*n* = 250 Cases)				*p*-Values for Interaction	
	Model 1		Model 2		Sex	*APOE* ε4
**Vitamins**	*HR* (95% CI)	*p*	*HR* (95% CI)	*p*		
Vitamin A (mg/L)	1.22 (0.72; 2.07)	0.453	1.22 (0.68; 2.18)	0.508	-	-
Beta-carotene (mg/L)	0.96 (0.64; 1.44)	0.850	1.02 (0.67; 1.57)	0.928	0.03	-
Vitamin E (mg/L)	1.01 (0.99; 1.03)	0.378	1.01 (0.99; 1.03)	0.500	-	-
Vitamin D (mmol/L)	0.99 (0.98; 0.99)	0.008	0.99 (0.98; 0.99)	0.015	-	-
**Vitamin D cut-offs**						
≥50 (mmol/L) (*n* = 449)	*Reference*		*Reference*			
≥25–<50 (mmol/L) (*n* = 548)	1.10 (0.80; 1.49)	0.568	1.19 (0.84; 1.67)	0.323		
<25 (mmol/L) (*n* = 337)	1.75 (1.28; 2.40)	0.001	1.91 (1.30; 2.81)	0.001		

Based on imputed data. Abbreviations: *APOE* ε4, apolipoprotein ε4; CI = confidence interval; HR, hazard ratio. Model 1 is adjusted for age, sex, *APOE* ε4 and education. Model 2 is adjusted as for model 1, plus BMI, physical activity, smoking, memory decline before study baseline, total cholesterol, creatinine, triglycerides, systolic blood pressure, diastolic blood pressure and vitamin supplement intake (in addition to vitamin D: vitamin D supplement intake and month of blood sampling). A *p*-value < 0.05 is considered to be statistically significant.

**Table 3 nutrients-15-00061-t003:** Longitudinal associations between blood (serum/plasma) concentrations of vitamins A, D and E and beta-carotene and incident AD over a 7-year follow-up period.

	Incidence of AD (*n* = 209 Cases)				*p*-Values for Interaction	
	Model 1		Model 2		Sex	*APOE* ε4
**Vitamins**	*HR* (95% CI)	*p*	*HR* (95% CI)	*p*		
Vitamin A (mg/L)	1.11 (0.61; 2.01)	0.735	1.02 (0.52; 2.00)	0.950	-	-
Beta-carotene (mg/L)	0.96 (0.61; 1.49)	0.838	0.98 (0.61; 1.57)	0.939	0.01	-
Vitamin E (mg/L)	1.01 (0.99; 1.03)	0.173	1.01 (0.99; 1.04)	0.270	-	-
Vitamin D (mmol/L)	0.98 (0.98; 0.99)	0.001	0.99 (0.98; 0.99)	0.003	-	-
**Vitamin D cut-offs**						
≥50 (mmol/L) (*n* = 449)	*Reference*		*Reference*			
≥25–<50 (mmol/L) (*n* = 548)	1.41 (0.99; 1.99)	0.056	1.52 (1.04; 2.22)	0.031		
<25 (mmol/L) (*n* = 337)	2.06 (1.44; 2.96)	<0.001	2.28 (1.47; 3.53)	<0.001		

Based on imputed data. Abbreviations: AD = Alzheimer’s dementia; *APOE* ε4, apolipoprotein ε4; CI = confidence interval; HR, hazard ratio. Model 1 is adjusted for age, sex, *APOE* ε4 and education. Model 2 is adjusted as for model 1, plus BMI, physical activity and smoking, memory decline before study baseline, total cholesterol, creatinine, triglycerides, systolic blood pressure, diastolic blood pressure and vitamin supplement intake (in addition to vitamin D: vitamin D supplement intake and month of blood draw). A *p*-value < 0.05 is considered to be statistically significant.

**Table 4 nutrients-15-00061-t004:** Analyses stratified by sex (*n* = 1334).

	Incidence All-Cause Dementia (*n* = 250 Cases)			
	Model 1		Model 2	
	HR (95% CI)	*p*	HR (95% CI)	*p*
**By sex**				
Beta-carotene (mg/L)				
Men (*n* = 473)	1.59 (0.79; 3.19)	0.195	1.32 (0.63; 2.77)	0.458
Women (*n* = 861)	0.81 (0.49; 1.34)	0.414	0.93 (0.55; 1.58)	0.790

*p* for interaction = 0.03. Based on imputed data. Abbreviations: *APOE* ε4, apolipoprotein ε4; AD = Alzheimer’s dementia; HR, hazard ratio; CI = confidence interval; *p*, *p*-value. Model 1 is adjusted for age, *APOE* ε4 and education. Model 2 is adjusted as for model 1, plus BMI, physical activity and smoking, memory decline before study baseline, total cholesterol, creatinine, triglycerides, systolic blood pressure and diastolic blood pressure and vitamin supplement intake. A *p*-value < 0.05 is considered to be statistically significant.

**Table 5 nutrients-15-00061-t005:** Analyses stratified by sex (*n* = 1334).

	Incidence of AD (*n* = 209 Cases)			
	Model 1		Model 2	
	HR (95% CI)	*p*	HR (95% CI)	*p*
**By sex**				
Beta-carotene (mg/L)				
Men (*n* = 473)	1.87 (0.92; 3.63)	0.086	1.33 (0.64; 2.78)	0.444
Women (*n* = 861)	0.73 (0.41; 1.31)	0.293	0.83 (0.45; 1.54)	0.555

*p* for interaction = 0.01. Based on imputed data. Abbreviations: *APOE* ε4, apolipoprotein ε4; AD = Alzheimer’s dementia; HR, hazard ratio; CI = confidence interval; *p*, *p*-value. Model 1 is adjusted for age, *APOE* ε4 and education. Model 2 is adjusted as for model 1, plus BMI, physical activity and smoking, memory decline before study baseline, total cholesterol, creatinine, triglycerides, systolic blood pressure and diastolic blood pressure and vitamin supplement intake. A *p*-value < 0.05 is considered to be statistically significant.

## Data Availability

Data described in the manuscript, code book, and analytic code will be made available to qualified researchers upon reasonable request.

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
