# Peer review of "Low Serum Vitamin D Status Is Associated with Incident Alzheimer’s Dementia in the Oldest Old"

_nutrients, 2022, doi:10.3390/nu15010061_

Round 1

Reviewer 1 Report

The paper’s title is focused on Vitamin D yet the intro provides information as E > A > D. L76 – why is Vitamin D more promising?

L111 – what were the exclusion criteria, only what’s listed (death, missing data etc.?).

L121 – were samples protected from light and heat?

L211- “gender” should be replaced with “sex.”

Data presentation as model1/2 is fine, however an additional graph showing concentrations from Y1 to Y7 should be included; specifically showing differences at Y1 for Vitamin D levels for people who developed dementia or AD vs cognitively normal. The presentation as is does show significance for Vitamin D, however it makes the data unclear if Vitamin D levels were lower in these patients at Y1 or if a drop in Vitamin D was only apparent at Y7. If so, then is it possible that Vitamin D deficiency is just occurs with AD rather than “increasing risk for AD” as described in the abstract?  

The format for the discussion is strange – why is it broken into subheadings like a review article? The section of oxidative damage is pretty vague and could be integrated into the other sections.

Author Response

We thank the reviewer for the positive feedback on our manuscript. Please find our response in the pdf.

Sincerely, 

Debora Melo van Lent

Reviewer 2 Report

Manusktryp concerns a very interesting study on the effect of the vitamin deficiency link on the occurrence of AD. As people live longer, dementia or AD will become an increasing social problem, making the search for factors to reduce the likelihood of their occurrence extremely important. However, the paper is missing some important information. The following is a list:

Lines 98-99 a brief description of patient inclusion and exclusion should be inserted. The mere fact that it is described in another publication does not give consideration to those who do not have access to the mentioned publications. 

The paper describes the analytical studies, but lacks information on the volume of the sample taken. What volume of sample was tested in each experiment. Were the procedures used validated? If so, where was this described and published. If not, then the procedures used are unreliable. 

If, on the other hand, validation was performed at the time of these studies, this must be described.

Missing from the patient information are those concerning the medications used by patients that can slow the progression of dementia, such as cerebral vasodilators. Or medications that reduce the absorption of the vitamins studied.

Author Response

We thank the reviewer for the positive feedback on our manuscript. Please found our response in the attached pdf.

Sincerely,

Debora Melo van Lent

Round 2

Reviewer 1 Report

Concerns addressed.